# A Single Session of Virtual Reality Improved Tiredness, Shortness of Breath, Anxiety, Depression and Well-Being in Hospitalized Individuals with COVID-19: A Randomized Clinical Trial

**DOI:** 10.3390/jpm12050829

**Published:** 2022-05-19

**Authors:** Isabele Moraes Rodrigues, Adriana Gomes Lima, Ana Evelyn dos Santos, Anne Carolline Almeida Santos, Luciana Silva do Nascimento, Maria Veronica Cavalcanti Lins Serra, Terezinha de Jesus Santos Pereira, Felipe Douglas Silva Barbosa, Valquiria Martins Seixas, Katia Monte-Silva, Kelly Regina Dias da Silva Scipioni, Daniel Marinho Cezar da Cruz, Daniele Piscitelli, Michela Goffredo, Miburge Bolivar Gois-Junior, Aristela de Freitas Zanona

**Affiliations:** 1Department of Occupational Therapy, Universidade Federal de Sergipe, Lagarto 49400-000, SE, Brazil; babi.moraes.r@gmail.com (I.M.R.); arisz_to@yahoo.com.br (A.d.F.Z.); 2Occupational Therapy Service, Hospital Universitário Lagarto, Lagarto 49400-000, SE, Brazil; adclala@gmail.com (A.G.L.); felipedouglas@live.com (F.D.S.B.); 3Occupational Therapy Service, Hospital Regional Dr. Jessé Fontes, Estância 49400-000, SE, Brazil; anaevelynsts@gmail.com; 4Occupational Therapy Service, Hospital de Urgencias de Sergipe, Aracaju 49095-000, SE, Brazil; karolline_almeida14@hotmail.com; 5Occupational Therapy Service, Hospital Getúlio Vargas, Recife 49095-000, PE, Brazil; nascimentoluciana@hotmail.com (L.S.d.N.); veronicaserra1@hotmail.com (M.V.C.L.S.); 6Occupational Therapy Service, Hospital Universitário Walter Cantidio HUWC, Fortaleza 49095-000, CE, Brazil; terezinhajsp@hotmail.com; 7Postgraduate Program in Applied Health Sciences (PPGCAS), Lagarto 49400-000, SE, Brazil; valquiria.mseixas@gmail.com; 8Department of Physical Therapy, Universidade Federal de Pernambuco, Recife 50670-901, PE, Brazil; monte.silva@ufpe.br; 9Department of Occupational Therapy, Universidade Federal do Paraná, Curitiba 80210-170, PR, Brazil; kellytoufpr@gmail.com; 10Occupational Therapy (Pre-Registration) Programme, School of Health, Leeds Beckett University, Leeds LS1 3HE, UK; D.M.Cezar-Da-Cruz@leedsbeckett.ac.uk; 11School of Medicine and Surgery, University of Milano Bicocca, 20126 Milano, Italy; daniele.piscitelli@unimib.it; 12School of Physical and Occupational Therapy, McGill University, Montreal, QC H3G 1Y5, Canada; 13Physical Therapy Program, Department of Kinesiology, University of Connecticut, Storrs, CT 06269, USA; 14Department of Neurological and Rehabilitation Sciences, IRCCS San Raffaele Roma, 00163 Rome, Italy; 15Laboratory of Motor Control and Body Balance (LCMEP), Center for Health Science, Universidade Federal de Sergipe, Aracaju 49400-000, SE, Brazil; miburgejr@hotmail.com

**Keywords:** COVID-19, virtual reality, well-being, symptom assessment, occupational therapy

## Abstract

Background: In 2020, the world was surprised by the spread and mass contamination of the new Coronavirus (COVID-19). COVID-19 produces symptoms ranging from a common cold to severe symptoms that can lead to death. Several strategies have been implemented to improve the well-being of patients during their hospitalization, and virtual reality (VR) has been used. However, whether patients hospitalized for COVID-19 can benefit from this intervention remains unclear. Therefore, this study aimed to investigate whether VR contributes to the control of pain symptoms, the sensation of dyspnea, perception of well-being, anxiety, and depression in patients hospitalized with COVID-19. Methods: A randomized, double-blind clinical trial was designed. Patients underwent a single session of VR and usual care. The experimental group (*n* = 22) received VR content to promote relaxation, distraction, and stress relief, whereas the control group (*n* = 22) received non-specific VR content. Results: The experimental group reported a significant decrease in tiredness, shortness of breath, anxiety, and an increase in the feeling of well-being, whereas the control group showed improvement only in the tiredness and anxiety. Conclusions: VR is a resource that may improve the symptoms of tiredness, shortness of breath, anxiety, and depression in patients hospitalized with COVID-19. Future studies should investigate the effect of multiple VR sessions on individuals with COVID-19.

## 1. Introduction

In 2020, the world was surprised by the mass spread and contamination of the new Coronavirus (COVID-19), leading to a pandemic, deaths, and post-COVID syndrome. The first official case of COVID-19 emerged in China in December 2019 [1] and quickly spread to other countries, leading the World Health Organization (WHO) on 30 January 2020, to declare a state of an international public health emergency [2]. Shortly, on 11 March 2020, the WHO declared the outbreak a global pandemic given the severity of the spread and the severity of the disease [3]. COVID-19 produces symptoms ranging from a fever, cough, tiredness, loss of taste or smell, sore throat, headache, pain and discomfort, and diarrhea, to severe symptoms such as loss of speech, mobility or confusion, and difficulty breathing or shortness of breath that can lead to death [4]. Thus, medical management depends on the individual’s symptomatic picture. Mild cases should be treated on an outpatient basis, with self-isolation and guidance for all house residents [5]. In severe cases in which the patient has breathing difficulties, hospitalization and even intubation are essential [1,5]

Isolation and hospitalization can lead to subjects’ withdrawal from their routine, generating anxiety [6,7,8], and psychiatric comorbidities [9]. Notably, patients may develop feelings of anguish about what will happen from that moment on [10]. Thus, strategies such as telerehabilitation, augmented reality, and serious games have been proposed to improve the patient’s quality of life and well-being to mitigate the impacts of hospitalization and COVID-19 [11].

With technological advances over the years, it has been possible to implement various technologies within the therapeutic process within hospitals. Healthcare professionals have sought alternatives that may be implemented within the inpatient population to assist and enhance their health recovery [12], considering the importance of biopsychosocial aspects related to the hospitalized person. Alternatives such as virtual visits, cinemas, and virtual reality (VR) have been implemented in hospital settings to increase the well-being of patients during their hospitalization. VR and exergames have also been recognized as effective and stimulating tools in neurological [13] and orthopedic [14] rehabilitation. Moreover, VR has been used as a tool for the control of pain and discomfort associated with burn care [15], for chronic pain [16], to reduce suffering during cancer treatment [17], and to improve pain during the debridement of burns in the hospital context [18]. VR provides realistic and three-dimensional experiences that promote users’ illusion of being in another environment [12]. Overall, this resource has shown to be promising for its low costs and positive results.

However, it is unclear whether hospitalized patients with COVID-19 can benefit from VR to reduce pain and improve dyspnea, anxiety attacks, and depression. Thus, this study aimed, through a randomized clinical trial, to investigate whether a single therapeutic VR session in the experiment group contributes to controlling pain symptoms, dyspnea, perception of well-being, anxiety, and depression compared to a non-therapeutic VR session in the control group in hospitalized patients with COVID-19. In addition, it is unclear whether VR can improve clinical signs such as heart rate, respiratory rate, blood oxygen saturation (SpO_2_), and blood pressure.

## 2. Materials and Methods

This is a multicenter, randomized, double-blind clinical trial following the recommendations of the CONSORT guideline [19].

The study followed the guidelines of Resolution 466/12 of the National Health Council (Brazil) and was conducted according to the Declaration of Helsinki. The research was approved by the Ethics Committee in Research with Human Beings of the Federal University of Sergipe, under CAAE no: 40165820.6.0000.5546.

All participants were informed of the objectives and procedures of the study. Before starting any related study procedures, written informed consent (Informed Consent Form, ICF) was obtained from the study participants.

### 2.1. Participants

Participants were enrolled from inpatient wards for patients with COVID-19 at four sites, i.e., (i) Dr. Jessé Fontes Regional Hospital in the municipality of Estância in the state of Sergipe/Brazil; (ii) University Hospital in the municipality of Lagarto in the state of Sergipe/Brazil; (iii) Hospital Getúlio Vargas of Recife in the state of Pernambuco/Brazil; and (iv) the Walter Cantídio University Hospital, in Fortaleza in the state of Ceará/Brazil. From December 2020 through July 2021, the research was carried out by occupational therapists trained in the assessment procedures and intervention protocol.

Individuals of all genders, between 18 and 80 years old, of all races and ethnicities, who tested positive for COVID-19 and were hospitalized participated in the research. The sample was selected for convenience.

### 2.2. Inclusion and Exclusion Criteria

Inclusion criteria were: individuals diagnosed with COVID-19, who may have complaints of pain, anxiety, malaise, dyspnea, using or not using oxygen, in a state of wakefulness and alertness, with a Mini-Mental State Examination–MMSE ≥ 18 [20,21].

Exclusion criteria were: subjects with labyrinthitis and/or vestibular disorder, history of motion sickness or nausea, neurological and psychiatric comorbidities, and receptive language impairments. Individuals with other injuries (e.g., total/complete visual impairment and total/complete hearing impairment) interfering with the experimental procedure and individuals who did not sign the ICF were excluded.

### 2.3. Study Closure or Discontinuity Criteria

Participants who (i) reported some discomfort during the intervention; (ii) withdrew from participating in the study during data collection; and (iii) did not allow the assessment or the use of VR glasses were excluded from the study.

To reduce the risk of respiratory distress, the occupational therapist kept the ventilation of the environment as pleasant as possible and took care not to occlude the patient’s nostrils during the procedure, so as not to make breathing difficult. If the patient was using some oxygen supplementation equipment, the occupational therapist observed whether it functioned normally. In the case of physical discomfort, the occupational therapist used positioning techniques and stopped the intervention.

The intervention was terminated in cases of risk or permanent damage to the health of the patient participating in the research, where this risk or damage was not foreseen in the ICF and in the cases of death, as well as in the cases where the patient gave up participating in the research.

As a primary outcome measure, we evaluated the perception of pain, dyspnea, anxiety, stress, depression, and well-being. As a secondary outcome, we evaluated arterial hypertension, heart rate, respiratory rate, and SpO_2_.

### 2.4. Primary Outcome Measures

#### 2.4.1. Edmonton Symptom Rating Scale

The Edmonton Symptom Rating Scale comprises visual numerical scales ranging from 0 to 10, where 0 represents the absence of the symptom or the best condition, and 10 represents the worst possible symptom. This assessment was designed to allow quantitative measurements of the intensity of symptoms presented by patients, with the option of adding a tenth symptom referring to what they are currently feeling [22,23].

#### 2.4.2. The Borg Scale for Perceived Effort

The Borg Scale for Perceived Effort is a one-dimensional instrument that assesses perceived exertion during physical activity [24]. The scale values range from 6, which means “no effort”; 9, corresponding to a “Very light” exercise; 13, corresponding to a “Slightly difficult” exercise; 17, corresponding to a “Very difficult” exercise; 19, corresponding to an exercise that is “Extremely difficult”; and 20, which means maximum effort (Figure 1).

#### 2.4.3. Hospital Anxiety and Depression Scale

The Hospital Anxiety and Depression Scale (HADS) is a reliable outcome measure for the screening of anxiety and depression [25]. This scale contains 14 Likert-based questions, of which 7 items are for anxiety and 7 for depression. The global score on each subscale ranges from 0 to 21 [26]. The level of depression is defined according to each subscale score, i.e., scores from 0 to 7, no anxiety and/or depression; scores from 8 to 10, suspected anxiety and/or depression; and scores of 11 or more indicates a possible case of anxiety and/or depression [25]. The score is calculated by summing the patient’s responses, where higher scores indicate higher anxiety and/or depression levels.

### 2.5. Secondary Outcome Measures

The following vital signs were measured: heart rate, respiratory rate, blood pressure, and SpO_2_, through monitors, and in cases of unmonitored patients, the values were collected using an oximeter and sphygmomanometer and were recorded on a medical record daily assessment. Length of stay and intubation rate were as recorded on the form after the session.

### 2.6. Randomization, Blinding, and Allocation

Patients were randomly allocated into two groups: the experimental and control groups. The allocation was concealed through the website www.random.org (accessed on 2 December 2020), in which, based on the patients’ recruitment, the participant received a number that the program randomly assigned to them, to place then in one of the two groups. All participants were blind about their group assignment.

In the experimental group, patients underwent the usual therapy, consisting of the following: activities to guide the hospitalization process, coping with the hospitalization process, energy conservation, Daily Living Activity training, Cognitive Rehabilitation, virtual call/visit, positioning, mobility joint, functional mobility, kinesiotherapy, assistive technology, sensory stimulation tailored on patient needs, combined with VR therapy. During VR, 360° videos were used with images of landscapes and/or mindfulness techniques presented to the patient to promote relaxation, distraction, and stress relief.

For the control group, patients underwent the usual therapy administrated by the Occupational Therapy unit of the hospital described above. Patients also used VR glasses; however, instead of therapeutic videos, they were exposed to a video with advertisements not related to relaxation and well-being content.

### 2.7. Study Procedures

In the initial screening, the Mini-Mental State Examination (MMSE) was applied (cutoff point of 18 points); the Katz Index and the case history were reviewed by the researchers that investigated the exclusion criteria. The Katz Index is an instrument for measuring the performance of six ADLs: Bathing, Dressing, Going to the Bathroom, Transference, Continence, and Feeding [27]. Each activity is scored on a dichotomous 0/1 scale, where 1 represents independence and 0 represents the need of supervision or assistance. Total scores range from 0 to 6, where higher scores represent the full function.

If the individual was able to participate in the research and agreed to participate, signing the ICF, the occupational therapist forwarded the patients’ name and his/her order number to the main researcher, who carried out the draw for the allocation of the participant in one of the groups. After that, the participant was placed in a comfortable position and the assessments were made, using The Borg Scale for Perceived Effort and Hospital Anxiety and Depression Scale. Then, the occupational therapist recorded the initial day, the values of the vital signs: heart rate, respiratory rate, blood pressure, and saturation on the daily assessment form, and filled out the Edmonton Symptom Assessment Scale. The glasses were placed on the participant after the evaluations were carried out, and the patient watched a 10-min video, according to the group assignment. After the intervention, the VR equipment was sanitized, and the battery of assessments applied in the pre-intervention was repeated after the intervention and the measurement of vital signs. Intervention time with virtual reality was 10 min, followed by 30 min of usual therapy for both groups.

### 2.8. Equipment

The equipment used in the study was the Oculos Realidade Virtual 3D Gamer Warrior JS080 (MULTILASER INDUSTRIAL SA, São Paulo, Brazil). The product is 16 cm high and 21.5 cm wide and supports all types of smartphones.

The equipment surface was sanitized using Lysoform and an alcohol-based lens cleaner was used for the glass lenses. We placed disposable film wraps on the VR goggles that were changed for each patient and placed them over the patient’s eyes to minimize direct contact with the device [12]. After the intervention, the equipment underwent a cleaning process before another patient could use it.

### 2.9. Data Analysis

Descriptive analyses were performed to depict the demographic and clinical characteristics of the two groups. Data are represented as frequency (with the relative percentage), mean value with Standard Deviation (SD), and median value with Interquartile range (IQR) for categorical, continuous, and ordinal variables, respectively.

For the clinical measurements, the nonparametric Wilcoxon test was used for within-group analyses, while the Mann–Whitney test was used for between-group analyses. Cohen’s d effect size was computed and interpreted as 0.2 small effect size, 0.5 medium size, and 0.8 large effect size.

Statistical analyses were performed using SPSS software (IBM SPSS Statistics for Windows, Version 20.0, IBM Corp, Armonk, NY, USA). A significance level of 5% (*p* ≤ 0.05) was adopted.

## 3. Results

A total of 44 patients allocated equally in both groups (22 participants in each group) were included (Figure 2).

At the baseline, the demographic and clinical characteristics (i.e., age, gender, MMSE, and Katz Index) were similar among the two groups. In the reported symptoms, only one subject reproduced pain and difficulties in ADL for each group (Table 1). Anxiety was reported in one subject in the experimental group and two subjects in the control group. Four participants reported shortness of breath in the experimental group and none in the control group. Feelings of depression, fear, and stress were reported in two participants in the experimental group and in three individuals in the control group. Three participants from both groups reported isolation/lack of family. In both groups, some participants did not report any symptoms (*n* = 10 in the experimental and *n* = 12 in the control group).

The within-group analysis showed a significant improvement in the Edmonton scale items on fatigue, shortness of breath, anxiety, and well-being in the experimental group. In contrast, the control group reported significant differences only in the fatigue and anxiety items. In the between-group analysis, no differences were found at the baseline between the control and experimental group. At the post-intervention, a significant difference was detected in favor of the experimental group for the well-being item (*p* = 0.045) (Table 2). There were no within-or between-group differences in perception of effort, as measured using the Borg Scale for Perceived Effort. Concerning the HADS scale, only the experimental group had a difference from baseline to post-intervention. No between-group differences were detected (see Table 2 for details).

The within-group analysis showed a significant improvement in the systolic blood pressure (*p* = 0.003) and heart rate (*p* = 0.003). There was no statistical difference in the other secondary outcomes. For both groups, there was an increase in oxygen saturation and a decrease in the respiratory rate in the within-group analysis (Table 3).

## 4. Discussion

Our results demonstrated a decrease in tiredness, shortness of breath, and anxiety, and an increase in the feeling of well-being in the experimental group, whereas the control group showed improvement only in the tiredness and anxiety items.

The result was similar to the study by Jung, et al. [28]. In their study, the VR was used for pulmonary rehabilitation in patients with chronic obstructive pulmonary disease, where participants reported improvement in dyspnea, fatigue, and emotional function after VR-assisted treatment. Fatigue and dyspnea derive from interactions between multiple physiological, psychological, social, and environmental factors and may induce secondary behavioral and physiological responses [29].

Addressing the influence of the sensory system on dyspnea [30] states that the sensory information involved in the breathing process is sent to higher brain centers, where signal processing modulates the expression of the evoked sensation, under the influence of cognitive and behavioral factors. The improvement in these aspects can be explained by the exposure of these patients to a relaxing virtual environment, “blocking” the stressful stimuli from the environment, thus evoking responses from the nervous system that fit the stimuli offered by VR.

In the within-group and between-group analysis, the well-being variable improved only for the experimental group. The VR may have improved well-being because it used a relaxing video, improving the parameters of tiredness, shortness of breath, and especially the emotional factor, as explained by [28]. This feeling of general well-being may have triggered the Autonomic Nervous System (ANS) to react to the given stimulus to alter vital signs. Notably, the ANS is responsible for coordinating the human body’s involuntary, visceral, and homeostatic functions, recognizing and responding to the environment and its changes [31]. In addition, the ANS is responsible for the body’s adaptations aimed at maintaining the body’s functioning and vital needs, being regulated by sensory feedback and closely related to experience and emotional expression [31]. Thus, depending on the patient’s state, whether relaxed or alert, the ANS will modulate vital signs, reacting according to the stimulus given; the more stressful the stimulus is, the more significant the response will be.

Thus, in our study, the improvement of the oxygen saturation parameters and the respiratory rate of the experimental group may have occurred due to immersion in a relaxing environment, offering participants the ability to control external stimuli, reducing the focus on the stressor factor, and, consequently, reorganizing respiratory responses. The fact that the control group showed positive results in reducing tiredness and anxiety, as well as improving the respiratory rate and oxygen saturation, can be explained by the positioning techniques, orientation to the hospitalization process, coping with the hospitalization process, conservation of energy, and exposure to VR, performed in both groups. We can also infer that the spontaneous improvement of the COVID-19 condition may have been responsible for improving the oxygen saturation and respiratory rate in both groups. However, despite both groups showing significant results, the experimental group had a slightly larger effect size for these outcomes, according to the effect size.

Epidemiologically, 20 to 60% of patients admitted to general hospitals suffer from psychiatric disorders, with depressive and anxiety disorders being the most frequent [32]. In the case of COVID-19, it was no different. In both assessments, patients in this study had high anxiety levels on the Edmonton scale and the HADS. On the HADS scale, this study showed that only the experimental group had a difference from baseline to post-intervention, a result similar to that of the study by [33]. In their randomized clinical trial aimed to evaluate the effectiveness of VR to support pulmonary rehabilitation, they reported a significant improvement in the levels of anxiety and depression evaluated with the HADS in the experimental group, thus indicating that VR was effective in this regard. The immersion of the patient in a relaxing environment through VR makes the levels of anxiety and depression fall as the patient becomes relaxed and consequently improves the mood and the feeling of well-being. During the therapy, there probably was a release of dopamine. This neurotransmitter regulates some emotions and is related to pleasure and well-being while using an electronic device [34], indicating a positive result in the sensation of the well-being of the experimental group in this study.

The low involvement in activities of daily living, both in the control and in the experimental group, can be explained by the fact that the hospital context can limit daily life activities such as bathing alone and changing postures. Some patients needed additional assistance as they had venous access devices and urinary catheters. Moreover, COVID-19 symptoms such as shortness of breath, tiredness, and fatigue may have limited independence in activities of daily living.

A few limitations should be discussed. Despite the large number of individuals hospitalized for COVID-19, we were not able to increase the sample size, as most of the patients admitted to the hospital were quickly sedated and intubated due to the severe condition they had when they were admitted. The present study was developed during Brazil’s first and second COVID-19 waves, from December 2020 through July 2021. Although we assessed more than 100 subjects (*n* = 102), the sample size was limited to 22 subjects per group (e.g., convenience sampling). Due to the relevant difficulties encountered during the pandemic (e.g., several patients required intubation and sedation for mechanical ventilation), we could not increase the sample size. Though a-priori power analysis was not run, and higher sample size would have provided a greater statistical power. The present study results are promising, revealing a positive effect (small-to-medium effect size) of VR in several domains. Further studies may consider using our findings to perform power calculations when developing randomized clinical trials on VR with COVID-19 individuals in similar environments. Additionally, cybersickness and technology acceptance were not assessed; however, none of the study participants reported discomfort, and both groups tolerated VR. Another limitation was the absence of a control group that did not receive the VR content. In the present study, the control group used the VR glasses without any relevant content; however, the fact that control subjects were wearing the glasses may have generated distraction and interfered with the results. Finally, future studies should investigate the effects of more VR sessions and determine the effects on activities of daily living in the hospital and own home environment.

## 5. Conclusions

This study aimed to investigate whether a single VR session contributes to controlling pain symptoms, dyspnea, perception of well-being, anxiety, and depression in hospitalized patients with COVID-19. Additionally, the contribution of VR to improving vital signs such as heart rate, respiratory rate, blood pressure, and SpO_2_, was evaluated.

Our study findings showed significant improvements in decreased tiredness, shortness of breath, anxiety, and an increase in the feeling of well-being in the experimental group, whereas the control group showed improvement only in the tiredness and anxiety items. Finally, the importance of this research is emphasized given the unprecedented situation of facing a disease that was, until recently, unknown by the population and the scientific community. This study favors that health professionals have scientific support for VR use with patients affected by COVID-19 to improve symptoms and outcomes. Future studies should investigate the effects of multiple VR sessions.

## Figures and Tables

**Figure 1 jpm-12-00829-f001:**
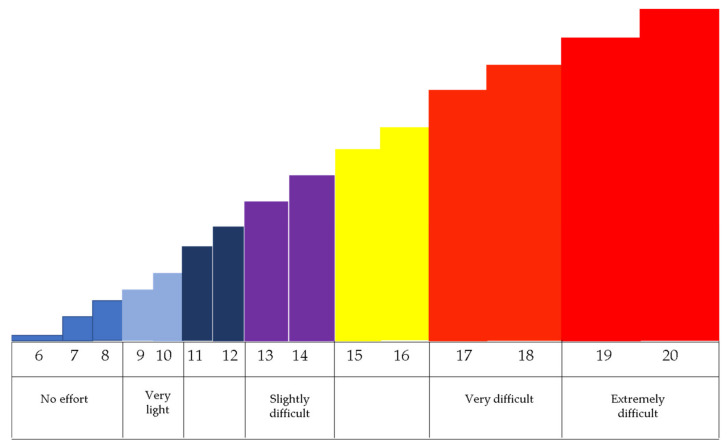
The Borg Scale for Perceived Effort. After the Virtual Reality session, the following question was asked to the participants: “How exhausting was this exercise?”.

**Figure 2 jpm-12-00829-f002:**
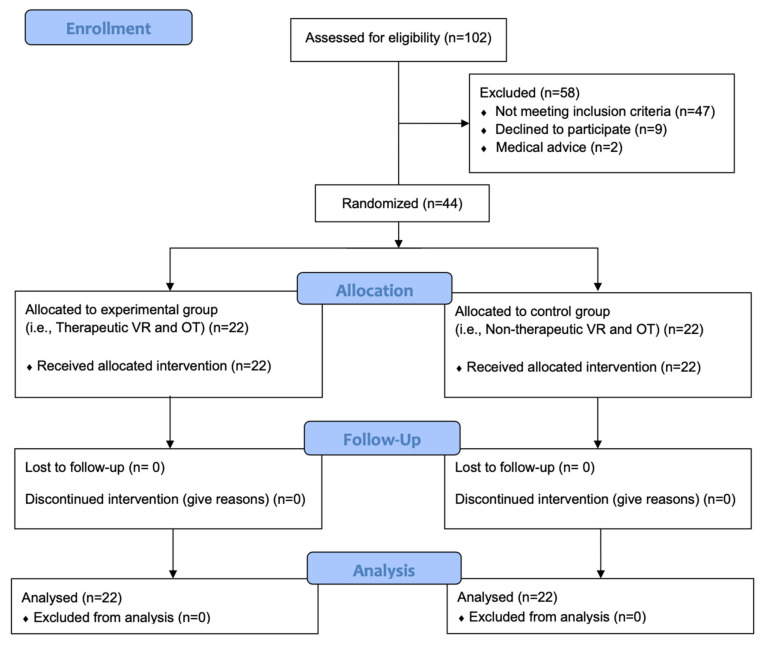
Consort Flow chart of the study.

**Table 1 jpm-12-00829-t001:** Demographic and clinical characteristics.

Characteristics	Experimental (*n* = 22)	Control (*n* = 22)	*p*-Value
Age (Mean ± SD)	48.9 ± 13.9	48.5 ± 16.9	0.998
Male gender *n* (%)	11 (50%)	11 (50%)	1.00
MMSE (Mean ± SD)	24.9 ±4	24 ± 3.8	0.668
Katz Index (Mean ± SD)	2.5 ± 2	2.8 ± 2.5	0.736

*n* = number; MMSE = Mini-Mental State Examination; SD = Standard Deviation.

**Table 2 jpm-12-00829-t002:** Primary outcomes.

Variables	Experimental Group				Control Group				EG vs. CG
	Baseline	Post Intervention	*p*-Value	Effect Size	Baseline	Post Intervention	*p*-Value	Effect Size	Effect Size Post vs. Post
ESRS									
Pain	1.20 (±1.78)	0.90 (±1.87)	0.394	0.164523	0.88 (±1.81)	0.59 (±1.24)	0.414	0.189524	0.19666
Weariness	2.60 (±3.09)	1.45 (±2.36) *	0.005	0.418701	1.35 (±1.81)	0.71 (±1.27) *	0.026	0.413114	0.39309
Somnolence	1.25 (±2.05)	0.95 (±2.27)	0.340	0.138861	1.71 (±2.02)	1.29 (±1.64)	0.084	0.223836	0.173964
Nausea	0.05 (±0.22)	0.00 (0.00)	0.317	-	0.29 (±0.75)	0.24 (±0.73)	0.655	0.079555	-
Appetite	1.15 (±2.13)	0.60 (±1.16)	0.066	0.321108	1.00 (±2.40)	0.65 (±1.97)	0.180	0.16076	0.029141
Shortness of breath	1.70 (±2.70)	1.15 (±2.15) *	0.026	0.225124	1.53 (±3.01)	0.65 (±1.53)	0.078	0.369275	0.269344
Depression	1.90 (±2.72)	1.90 (±2.88)	0.394	0.0	2.71 (±3.01)	2.35 (±2.76)	0.336	0.122245	0.16049
Anxiety	4.10 (±2.90)	2.10 (±2.23) *	0.001	0.773245	4.18 (±3.20)	2.94 (±2.55) *	0.011	0.426358	0.350552
Well-being	3.65 (±2.50)	2.30 (±2.79) *	0.029	0.50957	4.53 (±3.03)	4.18 (±2.87)	0.523	0.119475	0.661925 #
BSPE	12.13 (±4.55)	10.63 (±3.07)	0.063	0.386609	12.50 (±5.93)	12.38 (±5.93)	1.000	0.021089	0.370851
HADS	9.83 (±4.31)	7.17 (±2.79) *	0.042	0.734904	15.00 (±10.31)	13.00 (±9.49)	0.075	0.201883	0.000001

ESRS = Edmonton Symptom Rating Scale (note that 20 participants in the experimental group and 18 in the control group responded to ESRS, for the patient flow through the healthcare facility); BSPE = The Borg Scale for Perceived Effort; HADS Hospital Anxiety and Depression Scale; EG = Experimental Group; CG = Control Group; * *p* ≤ 0.05, within-group analysis baseline vs post intervention (Wilcoxon test); # *p* = 0.045, between-group analysis post with post intervention (Mann–Whitney U test).

**Table 3 jpm-12-00829-t003:** Secondary outcomes.

Variables	Experimental Group				Control Group				EG vs. CG
	Baseline	Post Intervention	*p*-Value	Effect Size	Baseline	Post Intervention	*p*-Value	Effect Size	Effect Size Post vs. Post
SBP (mmHg)	122.37 (±14.44)	119.32 (±13.76)	0.178	0.216249	128.24 (±22.01)	130.65 (±19.87)	0.780	0.114941	0.66295 #^a^
DPB (mmHg)	75.79 (±18.06)	73.47 (±17.80)	0.378	0.129389	74.12 (±28.24)	67.53 (±25.40)	0.056	0.245368	0.270841
SpO_2_ (%)	94.27 (±4.63)	95.64 (±3.58) *	0.016	0.331043	94.17 (±2.73)	94.94 (±2.69) *	0.018	0.284125	0.221069
HR (BPM)	86.95 (±15.34)	84.76 (±17.59)	0.080	0.1327	97.28 (±19.76)	96.33 (±19.04)	0.604	0.048961	0.631228 #^b^
RR (irpm)	26.41 (±9.89)	23.41 (±7.60) *	0.015	0.34015	23.50 (±6.32)	22.36 (±5.90) *	0.037	0.186469	0.154337

SBP = Systolic Blood Pressure; DBP = Diastolic Blood Pressure; SpO_2_ = blood oxygen saturation; HR = Heart Rate; RR = Respiratory Rate; EG = Experimental Group; CG = Control Group. * *p* ≤ 0.05. #^a^ *p* = 0.003 and #^b^ *p* = 0.001, between-group analysis post with post intervention (Mann–Whitney U test).

## Data Availability

The data that support the findings of the current study are available from the corresponding author [M.G.] upon reasonable request.

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
