# Peer review of "A Single Session of Virtual Reality Improved Tiredness, Shortness of Breath, Anxiety, Depression and Well-Being in Hospitalized Individuals with COVID-19: A Randomized Clinical Trial"

_jpm, 2022, doi:10.3390/jpm12050829_

Round 1

Reviewer 1 Report

The manuscript is well written and relatively easy to read and comprehend. 

I recommend changing lines 150 & 151 and other references to the Borg Scale as "The Borg Scale for Perceived Effort" and would suggest the addition of a figure to show the simplicity of the scale. 

On what basis did you settle on 22 subjects/group?  

Did you run Power Analysis a-priori?

Line 219: Should read Data Analysis

Line 232: Please rewrite. 

Page 6: The flow chart (figure 1) is not easy to follow.  Perhaps it's just the way it appears in the manuscript, but I think it can be simplified. 

Line 253,254: The Borg Scale is not a test.  I would suggest rewording this sentence: "There were no within-or between-group differences in perception of effort, as measured using the Borg Scale."  

Line 350: Change "Covid-19" to "COVID-19"

Line 4: Ditto

Lines 55 & 56: Ditto

Author Response

Please find attached the revised version of our paper entitled: “A single session of virtual reality improved tiredness, shortness of breath, anxiety, depression and well-being in hospitalized individuals with COVID-19: a randomized clinical trial” for publication on the Special Issue "Personalized Medicine for Covid-19 Patients-Clinical Considerations" of Journal of Personalized Medicine as “original article”.

We would like to take this opportunity to express our sincere thanks to the reviewers who identified fallacies in our manuscript that needed corrections or modification.

We addressed all the comments raised by the  Reviewers. Please find below our point-by-point responses. Changes were marked using the “Track  Changes” function in MS Word.

Thank you for your kind attention.

Yours sincerely,

the authors

Reviewer 2 Report

Thank you very much for giving me the opportunity to review the present manuscript entitled “A single session of virtual reality reduces pain, respiratory discomfort, anxiety, and depression in hospitalized individuals with Covid-19: a randomized clinical trial.”The authors investigated whether a single VR session contributes to controlling pain symptoms, dyspnea, perception of well-being, anxiety, and depression in hospitalized patients with COVID-19. The authors also evaluated the contribution of VR to improving vital signs such as heart rate, respiratory rate, blood pressure, and SpO2. The authors concluded that VR is a resource that may improve the symptoms of tiredness, shortness of breath, anxiety, and depression in patients hospitalized with COVID-19. The study initiatives were novel; the design was delicate; the selection of measurement tools and statistical methods were deliberate and suitable where appropriate. However, the methods were not robust enough, and result might not be contributable enough as expected. In addition, the preparation of this manuscript was not dedicated and had several errors.

My comments are listed below:

  1. The authors allocated randomly the patients into experiment group and control group. In the experimental group, patients underwent needed occupation therapy, combined with VR therapy. During VR, 360º videos were used with images of landscapes and/or mindfulness techniques presented to the patient to promote relaxation, distraction, and stress relief. For the control group, the participant underwent the usual therapy administrated by the Occupational Therapy unit of the hospital, and also used VR glasses as the experiment group, but were exposed to a video with advertisements not related to relaxation and well-being content. In this study design a “therapeutic VR” session was used in experiment group, and a “non-therapeutic VR” was used in control group. This was different from the original statement of objective of study where “a single session of VR” was used as an intervention. I suggest the authors may revise the statement of aim of study to match their study design.
  2. In a clinical trial, the primary analysis usually focuses on the comparison between experiment group and the control group (between-group analysis). But the authors put more content in presenting the results of within-group analysis, and then compare the within-group analysis between experiment and control group. Other outcome measurements such as differences between the pre- and post- session Edmonton scale, may provide more direct comparison between the two groups.
  3. The authors included the participants in a state of wakefulness and alertness, with a Mini-Mental State Examination – MMSE ≥ 18. The Mini-Mental State Examination (MMSE) is a good tool commonly used to check for cognitive impairment. A score of 20 to 24 suggests mild dementia, 13 to 20 suggests moderate dementia, and less than 12 indicates severe dementia. It is not clear why the authors chose 18 as a cut-off point? Was there any reference to support this decision?
  4. A major drawback of this study was the small sample size. Almost one half of the eligible patients were excluded according to exclusion criteria, and additional 11 patients were excluded for reasons of declining to participate and others. That less than half of eligible patients finally participated in the study might increase the chance of selection bias. In addition, a small case number might have increased standard deviation, and decreased the effect size. The effect sizes of the results were largely small to medium, or at most above medium, indicating limited clinical significances.
  5. Although the baseline Katz index between the experiment group and the control group were similar, they had the mean value of only 2.5 and 2.8, respectively. These low values of Katz index indicated low activity of daily life of the participants, which may need explanation in the Discussions section.
  6. In Figure 1, the experiment group and the control group were expected to appear in the left arm and the right arm respectively after patient allocation. But the descriptions of these two arms were totally identical (“Allocated to intervention”). Was it an error in preparation of this manuscript?
  7. In legend of Figure 2, the author did not state clearly which attribute’s Edmonton scale was presented in this Figure. This might confuse the readers or make a misleading image that a summation result of all attributes was demonstrated. Actually this Figure demonstrated only the result of well-being, which could only be realized from the statement in Results section. In addition, the chart of pre-session evaluation in control group had only 21 dots; was it another error in manuscript preparation?
  8. The Results showed VR session improved the symptoms of tiredness, shortness of breath, anxiety, and depression in experiment group. The between group comparison showed only improvement in well-being. The title “A single session of virtual reality reduces pain, respiratory discomfort, anxiety, and depression…” did not match these results, and needed to be revised.
  9. Regarding the references, reference 3 was a publication that cited declaration of World Health Organization (WHO), not from a publication from WHO. This usage of citations was against the common consensus in a strictly written manuscript. The topic of reference 9 “Years of life lost to COVID-387 19 in 81 countries” also did not match the statement “disruption of daily life” that cited this reference in Introduction.

Author Response

(The authors gave the same response as above.)

Round 2

Reviewer 2 Report

I appreciate the authors' efforts to revise this manuscript. They make present form of manuscript more clear and more readable. The statement of title discussions were more reasonable and match their findings in results.